RESEARCH ARTICLE  

# A filter at the entrance of the Golgi that selects vesicles according to size and bulk lipid composition

Maud Magdeleine[1], Romain Gautier[1], Pierre Gounon[2], Hélène Barelli[1], Stefano Vanni[1], Bruno Antonny[1]*

[1]CNRS, Institut de Pharmacologie Moléculaire et Cellulaire, Université Côte d'Azur, Valbonne, France; [2]Centre Commun de Microscopie Appliquée, Université Côte d'Azur, Nice, France

**Abstract** When small phosphatidylcholine liposomes are added to perforated cells, they bind preferentially to the Golgi suggesting an exceptional avidity of this organelle for curved membranes without stereospecific interactions. We show that the *cis* golgin GMAP-210 accounts for this property. First, the liposome tethering properties of the Golgi resembles that of the amphipathic lipid-packing sensor (ALPS) motif of GMAP-210: both preferred small (radius < 40 nm) liposomes made of monounsaturated but not saturated lipids. Second, reducing GMAP-210 levels or redirecting its ALPS motif to mitochondria decreased liposome capture by the Golgi. Extensive mutagenesis analysis suggests that GMAP-210 tethers authentic transport vesicles *via* the same mechanism whereby the ALPS motif senses lipid-packing defects at the vesicle surface through its regularly spaced hydrophobic residues. We conclude that the Golgi uses GMAP-210 as a filter to select transport vesicles according to their size and bulk lipid composition.

*For correspondence: antonny@ipmc.cnrs.fr

**Competing interests:** The authors declare that no competing interests exist.

## Introduction

Exchange of material between organelles is largely mediated by transport vesicles, which bud from the surface of a donor compartment through the mechanical action of protein coats and fuse with an acceptor compartment through the action of SNARE proteins (*Bonifacino and Glick, 2004*). Between these two elementary steps, various protein machineries control the fate of transport vesicles. These include molecular motors for long-range transport and tethering complexes, which trap vesicles in the vicinity of their destination organelle (*Cai et al., 2007*).

Golgins are long coiled-coil proteins at the surface of the Golgi apparatus. Through their string-like structure whose length might approach 300 nm, they have been proposed to form a matrix to tether transport vesicles in the vicinity of Golgi cisternae, while excluding some large particles such as ribosomes (*Malsam et al., 2005*; *Brown and Pfeffer, 2010*; *Munro, 2011*; *Witkos and Lowe, 2015*; *Gillingham and Munro, 2016*). In addition, recent work based on the artificial attachment of golgins to mitochondria indicates that golgins provide a first layer of vesicle selectivity (*Wong and Munro, 2014*). Golgins that are normally anchored on the *cis* side of the Golgi preferentially capture vesicles trafficking between the endoplasmic reticulum (ER) and the *cis* Golgi, whereas those that are normally anchored at the *trans* side preferentially capture vesicles from endosomal sources. Although the molecular determinants responsible for this selectivity are not well known, discrete binding sites for Rab proteins have been identified along the coiled-coils of golgins, suggesting a mechanism by which vesicles move within the golgin matrix by transient Rab-coiled-coil contacts (*Sinka et al., 2008*). EM and atomic force microscopy show that the coiled-coil architecture of

golgins is interrupted by regions of high flexibility, which might facilitate the movement of trapped vesicles (*Cheung et al., 2015*; *Ishida et al., 2015*).

Stereospecific interactions between proteins and lipids have an undisputed role in membrane traffic. However, other factors related to bulk physicochemical properties of membranes such as membrane curvature, electrostatics and lipid packing also play a role (*Bigay and Antonny, 2012*). In the case of vesicle tethering at the Golgi, Kobayashi and Pagano reported in 1988 a puzzling observation (*Kobayashi and Pagano, 1988*). When liposomes made of pure lipids were incubated with perforated cells, they bound almost exclusively to the Golgi and not to any other organelle. Binding was fast, occurred at 4°C, did not require ATP, and was observed with simple lipid compositions such as pure egg phosphatidylcholine (PC). In the presence of ATP and at temperature above 15°C, liposome binding was followed by membrane fusion and delivery of the liposome content to the Golgi. Interestingly, the authors noticed that liposomes with a small radius (R < 40 nm) were targeted more efficiently to the Golgi than larger ones (*Kobayashi and Pagano, 1988*). Together, these observations suggest that the Golgi has a superior ability to capture small vesicles than any other organelle because of an unexplained avidity for curved lipid membranes.

We previously presented a minimal model for vesicle tethering by the golgin GMAP-210, in which non stereospecific protein-membrane interactions play a predominant role (*Drin et al., 2008*). GMAP-210, which controls vesicle traffic at the *cis* side of the Golgi (*Roboti et al., 2015*), contains an N-terminal ALPS motif [1–38], several long coiled-coil regions, a GRAB domain [1774–1843], and a region of low complexity [1844–1979]. ALPS motifs are amphipathic helices that adsorb preferentially onto highly curved membranes containing unsaturated lipids (*Drin et al., 2007*; *Drin and Antonny, 2010*; *Antonny, 2011*). The GRAB domain interacts with membrane-bound Arf1-GTP (*Gillingham et al., 2004*). Because the Arf1 regulator ArfGAP1, which hydrolyzes GTP on Arf1, also contains an ALPS motif, the interaction between the GRAB domain and Arf1-GTP is not stable on curved membranes (*Drin et al., 2008*). Consequently, GMAP-210 tethers small liposomes through its ALPS motif to flatter membranes through its GRAB domain, but other geometrical combinations are not possible (*Drin et al., 2008*). In a cellular context, this model suggests that GMAP-210 connects transport vesicles to Golgi cisternae on the basis of their contrasting membrane curvature.

Despite some predictions of this model have been successfully tested in cells, notably the relative contribution of the ALPS motif and of the GRAB domain to vesicle *versus* cisternae binding (*Cardenas et al., 2009*; *Sato et al., 2015*), several major questions remain. First, does GMAP-210 account for the striking avidity of the Golgi for small liposomes? In other words, is GMAP-210 a major vesicle trap? Second, are the amphipathic properties of the ALPS motif of GMAP-210 sufficient to explain the selective tethering of the vesicles of the early secretory pathway at the cis-Golgi as opposed to other transport vesicles?

In this study, we first connect the avidity of the Golgi for small liposomes with the biochemical properties of the golgin GMAP-210. Second, we interrogate the binding mode of the ALPS motif of GMAP-210 in a cellular context by extensive mutagenesis. We observe that the ALPS motif is very permissive to mutations that should disrupt stereospecific interactions, but is supersensitive to mutations that modify its amphipathic character. Notably, the sparse distribution of hydrophobic residues along the ALPS sequence is key for vesicle recognition. We conclude that the ALPS motif of GMAP-210 acts as a filter at the entrance of the *cis* Golgi to select transport vesicles of the early secretory pathway based on their small size and high content in unsaturated lipids.

## Results

### Avidity of the Golgi for small liposomes containing monounsaturated lipids

The Kobayashi-Pagano experiment consists of peeling the upper membrane of cultured cells with a filter followed by incubating the resulting open cells with synthetic fluorescent liposomes (*Kobayashi and Pagano, 1988*). After several washes with medium, the cells are examined with a fluorescent microscope (*Figure 1A,B*). In pilot experiments, we observed the same effects as those previously reported. Liposomes made of PC bound almost exclusively to the Golgi apparatus. In addition, the extent of binding was higher with small liposomes prepared by sonication than with larger liposomes prepared by extrusion (*Figure 1B*).

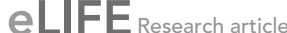

**Figure 1.** Preference of the Golgi for small unsaturated liposomes and biochemical properties of GMAP-210 ALPS motif. (A) Principles of the Kobayashi-Pagano experiment. (B) Typical observations. RPE1 cells stably expressing Rab6-GFP were perforated and incubated with DOPC liposomes (labeled with 1 mol% Rhod-PE). The liposomes were prepared by sonication or by extrusion through 200 nm (diameter) filters. The actual liposome

*Figure 1 continued on next page*

*Figure 1 continued*

radius (Rh) is indicated. Images were acquired with an epifluorescence microscope. (C, D) Experiments similar to that shown in (B) with liposomes of defined composition (C) or size (D). In (C) all liposomes were obtained by extrusion through 30 nm filters. In (D) all liposomes were used at a concentration of 250 μM lipids. The liposome fluorescence intensity in the Golgi area was determined. Each point corresponds to one cell from two to three independent experiments. (E) Domain organization of GMAP-210 and minimal model for membrane tethering (*Drin et al., 2008*). (F) Effects of membrane curvature and lipid unsaturation on the binding of the N-ter region of GMAP-210 to liposomes. Liposome binding was determined by monitoring the NBD fluorescence intensity. The plot shows the effect of liposome size and unsaturation from two independent experiments (triangles and inverted triangles). The fluorescence in solution was set at 1. Bars: 10 μm.

The following figure supplements are available for figure 1:

**Figure supplement 1.** Liposome tethering properties of perforated cells.

**Figure supplement 2.** Cumulative effects of membrane curvature and lipid unsaturation on the binding of the N-ter region of GMAP-210 to liposomes.

To perform a more comprehensive and quantitative analysis, we applied increasing amounts of liposomes of defined size and composition to perforated cells and established saturation curves (*Figure 1C,D* and *Figure 1—figure supplement 1*). The liposomes contained trace amounts of a red fluorescent lipid (Rhod-PE), whereas the cells stably expressed Rab6-GFP in order to visualize the Golgi area (*Figure 1B*). All experiments were performed with no ATP in the medium to minimize liposome fusion. We observed that liposome binding increased strongly with the unsaturation level of PC. Binding was maximal with PC containing two unsaturated acyl chains (C18:1-C18:1-PC = DOPC), barely detectable with PC containing two saturated acyl chains (C14:0-C14:0-PC = DMPC), and intermediate with PC containing one saturated and one monounsaturated acyl chain (C16:0-C18:1-PC = POPC) (*Figure 1C*). Moreover, liposome size had a strong effect on Golgi binding (*Figure 1D*). Below 50 nm in radius, liposomes bound significantly to the Golgi; above this radius, liposome binding was close to background levels. By calibrating the fluorescence intensity as a function of liposome concentration, we estimated that hundreds of small DOPC liposomes were trapped in the Golgi region of each perforated cell (see Materials and methods).

## GMAP-210 contributes to the liposome avidity of the Golgi

The cumulative effects of membrane curvature and lipid unsaturation on the binding of liposomes to the Golgi was reminiscent of the biochemical properties of ALPS motifs, which are present in various proteins (*Bigay et al., 2005*; *Drin et al., 2007*; *Vanni et al., 2014*). In the case of GMAP-210, the ALPS motif is present at the N-terminus of the golgin, which makes it ideally positioned to capture small transport vesicles or, in the case of perforated cells, small liposomes (*Figure 1E*).

The membrane binding properties of the ALPS motif of GMAP-210 have not been characterized as thoroughly as those of the ALPS motifs of ArfGAP1 (*Bigay et al., 2005*; *Mesmin et al., 2007*; *Vanni et al., 2014*). We used a NBD-labeled form of the N-terminal region [1–375] of GMAP-210 (*Pranke et al., 2011*) and conducted binding experiments with liposomes similar to those used on perforated cells except that they lacked fluorescent lipids. The increase in NBD fluorescence resulting from the adsorption of the ALPS motif onto the liposome surface was used as an index of membrane binding (*Drin et al., 2007*; *Pranke et al., 2011*). The N-terminal region of GMAP-210 showed a sharp preference for liposomes combining high curvature and high content of monounsaturated lipids (*Figure 1F* and *Figure 1—figure supplement 2*).

Given the good match between the liposome-binding properties of the Golgi and that of GMAP-210 in vitro, we surmised that GMAP-210 could contribute to the liposome trapping properties of the Golgi. To test this hypothesis, we reduced GMAP-210 expression using two recently described siRNA (*Roboti et al., 2015*). With both siRNA, the amount of GMAP-210 decreased ≈ seven-fold whereas the liposome tethering capacity of the Golgi decreased 2.5-fold (*Figure 2A* and *Figure 2—figure supplement 1*). In addition, we characterized the perforated cells by immunofluorescence. We observed good colocalization between the trapped liposomes and anti GMAP-210 or anti GM130 antibodies (*Figure 2B*). After treatment with siRNA against GMAP-210, the cells that showed background levels of GMAP-210 showed barely any significant liposome staining at the

**Figure 2.** GMAP-210 is responsible for the ability of the Golgi to capture liposomes. (**A**) RPE1 cells were treated with siRNA against GMAP-210 or with control siRNA. After 72 hr, the cells were perforated and small DOPC liposomes (250 μM lipids), which were obtained by extrusion through 30 nm

*Figure 2 continued on next page*

*Figure 2 continued*
polycarbonate filters, were applied. Liposome capture was quantified as in *Figure 1*. The western blot shows the amount of GMAP-210 in the siRNA-treated cells. (**B**) Immunofluorescence characterization of RPE1 cells after treatment with anti GMAP-210 siRNA or control siRNA followed by perforation and incubation with small DOPC liposomes obtained by extrusion. The actual liposome radius is indicated. The images show one Z section as obtained in a confocal microscope. The profiles show the fluorescence intensity of the three markers along the horizontal lines as indicated in the merged images. Bars: 10 µm. Error bars: SEM (n = 4)

The following figure supplement is available for figure 2:

**Figure supplement 1.** GMAP-210 is responsible for the ability of the Golgi to capture liposomes.

Golgi (*Figure 2B*). Altogether, the experiments shown in *Figure 2* indicate that GMAP-210 makes a major contribution to the liposome capture property of the Golgi.

To further assess the role of GMAP-210 in the liposome tethering properties of the Golgi, we combined a golgin derouting strategy (*Wong and Munro, 2014*) with the use of perforated cells (*Kobayashi and Pagano, 1988*). The cells expressed two constructs (*Figure 3A*). The first contained the following regions: the ALPS motif of GMAP-210 [1–38] to capture small liposomes, an artificial coiled-coil (ACC1) to impose a dimeric architecture (*Horchani et al., 2014*), a green fluorescent protein (GFP) for visualization, and a FK binding protein (FKBP) for control by rapamycin. The second construct, Mito-FRB, enabled the mitochondria attachment of the first construct by rapamycin-induced dimerization (*Wong and Munro, 2014*; *Sato et al., 2015*). In addition, we treated the cells with nocodazole, which, by disrupting the microtubules, causes the Golgi to disperse into mini stacks in the cytoplasm. This condition was chosen to increase the probability of liposome exchange between the mitochondria and the Golgi as observed in the case of authentic transport vesicles (*Wong and Munro, 2014*; *Sato et al., 2015*). After treatment with nocodazole and with or without rapamycin, the cells were perforated and DOPC liposomes labeled with Rhod-PE were added. Without rapamycin, the ALPS-containing construct and the small liposomes colocalized at Golgi mini stacks as identified by GM130 antibody (*Figure 3B*). With rapamycin, colocalization between the ALPS-containing construct and the liposomes remained, but the liposomes and the ALPS construct overlapped better with mitochondria than with the Golgi mini stacks (*Figure 3C* and *Figure 3—figure supplement 1*). Furthermore, only small liposomes but not large liposomes bound to mitochondria through the ALPS-containing construct (*Figure 3—figure supplement 2*). Thus, mitochondria-anchored ALPS efficiently competes with the Golgi to capture small liposomes.

## Selective *cis* Golgi targeting of the ALPS motif of GMAP-210

Having determined the contribution of GMAP-210 to the exceptional liposome tethering activity of the Golgi, we next addressed the mechanism of recognition of authentic transport vesicles. Recent works showed that GMAP-210 selectively captures vesicles of the early secretory pathway, thereby contributing to efficient forward and backward traffic between the ER and the Golgi (*Wong and Munro, 2014*; *Roboti et al., 2015*). Furthermore, deletion of the ALPS motif abolishes this activity (*Sato et al., 2015*). We wished to determine whether the ALPS motif of GMAP-210 recognizes transport vesicles through the same mechanism as that observed in vitro, that is, *via* its amphipathic properties (*Drin et al., 2007*; *Pranke et al., 2011*). Given the high selectivity of GMAP-210 for vesicles of the early secretory pathway (*Wong and Munro, 2014*; *Roboti et al., 2015*), an alternative possibility was that the ALPS motif engages in stereospecific interactions with a protein or a lipid.

We designed various fluorescent constructs derived from the N-terminal region of GMAP-210 (*Figure 4A*). ALPS-GCC-GFP (or mCherry) contained the N-terminal ALPS motif [1–38] and one quarter of the downstream coiled-coil region [39–375] (GCC) of GMAP-210, which was C-terminally fused to GFP (or mCherry). The other constructs were based on the same tripartite architecture (sensor/coiled-coil/fluorescent reporter), but contained a mutated ALPS motif and/or an artificial coiled-coil sequence (ACC1 or ACC2) to address the role of these regions in Golgi localization. The design of the artificial coiled-coil sequences ACC1 and ACC2 has been recently presented (*Horchani et al., 2014*). Note that none of the constructs contained the GRAB domain. Consequently, they could not target Golgi cisternae by interacting with Arf1-GTP (*Cardenas et al., 2009*; *Gillingham et al., 2004*) and should act as vesicle sensors rather than as vesicle tethers.



**Figure 3.** Mitochondria-anchored GMAP-210 ALPS competes with Golgi mini stacks for the capture of small liposomes. (**A**) Scheme of the constructs and principle of the experiment. Cells expressing ALPS-ACC1-GFP-FKBP and Mito-FRB were treated with nocodazole to disperse the Golgi apparatus into mini stacks. Thereafter, the cells were either treated with rapamycin or not, perforated, and incubated with small DOPC liposomes (extrusion: 30 nm). (**B**) In the absence rapamycin, small liposomes (extrusion 30 nm) colocalize with ALPS-ACC1-GFP-FKBP and with the Golgi mini stacks (as stained by anti GM130 antibody). (**C**) In the presence rapamycin, the small liposomes and ALPS-ACC1-GFP-FKBP colocalize in many regions that are negative for GM130. All images were acquired with a confocal microscope and were analyzed using Volocity software. These experiments were repeated three times. Data show representative images. Scale bars: 10 μm.

The following figure supplements are available for figure 3:

**Figure supplement 1.** Competition between mitochondria-anchored GMAP-210 ALPS and Golgi mini stacks for liposome capture.

**Figure supplement 2.** Mitochondria-anchored ALPS of GMAP-210 captures small but not large liposomes.



**Figure 4.** The ALPS motif of GMAP-210 targets *cis* Golgi vesicles. (A) Scheme of the constructs. The first construct corresponds to the N-terminal region of GMAP-210 (aa 1–375; ALPS = 1–38) fused to a C-ter fluorophore (GFP or mCherry). Alternatively, the ALPS motif of GMAP-210 was introduced upstream of an artificial coiled-coil (ACC1). (B) Subcellular localization of GCC- and ACC-based constructs by confocal fluorescence microscopy. Golgi localization was preserved by replacing the coiled-coil region of GMAP-210 (GCC) by an artificial coiled-coil (ACC1), but was abolished by deletion of the ALPS motif. Immunofluorescence of cells expressing ALPS-ACC1-GFP shows low colocalization with TGN46, intermediate overlap with GM130, and high colocalization with endogenous GMAP-210. All images were acquired with a confocal microscope and were analyzed using Volocity software. Scale bars: 10 μm, Z plane. (C) Immunogold EM of RPE1 cells expressing ALPS-ACC1-GFP. Scale bars: 10 μm (B) 100 nm (C). These experiments were repeated at least three times, except EM, which was repeated twice. Data show representative images.

The following figure supplements are available for figure 4:

**Figure supplement 1.** GMAP-210 depletion by siRNA does not affect the Golgi targeting of ALPS-containing probes.

**Figure supplement 2.** ALPS-ACC1-GFP increases the liposome capture properties of cells.

When coexpressed in RPE1 cells, ALPS-GCC-GFP and ALPS-ACC1-mCherry perfectly co-localized to a perinuclear region (*Figure 4B*). In contrast, ACC1-GFP, which lacks the ALPS motif, was entirely soluble (*Figure 4B*). Thus, the ALPS motif of GMAP-210 is not only necessary (as revealed by its deletion), but also sufficient (as revealed by coiled-coil permutation) for cellular localization. Double

labeling with Golgi markers and with endogenous GMAP-210 indicated that ALPS-containing constructs display a *cis* (GM130) rather than a *trans* (TGN46) localization (*Figure 4B*). However, colocalization with GM130 was partial, in agreement with the observation that GMAP-210 is enriched in *cis* Golgi regions from which GM130 is absent (*Cardenas et al., 2009*). Immunogold EM indicated that ALPS-ACC1-GFP labeled regions very rich in small (R ≈ 30 nm) vesicles (*Figure 4C*).

Next, we assessed the effect of reducing GMAP-210 levels on the targeting of ALPS-containing constructs to the Golgi. In contrast to what we observed for the capture of liposomes (*Figure 2*), cells treated with siRNA against GMAP-210 kept robust Golgi staining of ALPS-ACC1-GFP as well as with catalytically inactive ArfGAP1-GFP, another ALPS-containing protein (*Vanni et al., 2014*) (*Figure 4—figure supplement 1*). This result can be interpreted in light of previous EM observations. GMAP-210 depletion causes small vesicles to accumulate at the Golgi at the expense of the normal cisternae organization (*Sato et al., 2015*), an effect that probably results from a reduction in speed of Golgi assembly through homotypic vesicle fusion. Therefore, whereas GMAP-210 depletion reduces the possibility of small liposomes to interact with the Golgi, it still preserves the ability of ALPS-motif containing proteins to target this organelle *via* contact with authentic transport vesicles.

As an alternative strategy to connect the liposome capture properties of the Golgi and the behavior of the ALPS-ACC1-GFP probe, we added small liposomes on perforated cells overexpressing ALPS-ACC1-GFP. As shown in *Figure 4—figure supplement 2*, cells expressing ALPS-ACC1-GFP showed higher levels of liposome capture than neighboring cells that do not express this construct. Together with the effect of GMAP-210 depletion (*Figure 2*), this observation further underlines the contribution of the ALPS motif of GMAP-210 to the ability of the Golgi to capture small liposomes.

## The ALPS motif of GMAP-210 does not engage stereospecific interactions

To test whether the fine Golgi localization conferred by GMAP-210 ALPS motif relied on stereospecific interactions, we inverted the ALPS sequence (i.e. read it from the carboxy terminus to the amino terminus; *Figure 5A*). A helical plot shows that this inversion keeps intact the physicochemical features of the ALPS motif including its amphipatic character and, therefore, should not affect its membrane adsorption properties (*Figure 5B*). In contrast, the inversion corresponded to 30 mutations over 38 positions (*Figure 5A*) and should strongly perturb any stereospecific interaction with a particular protein or lipid. We previously showed that inverting the ALPS sequence does not compromise the Golgi localization of GMAP-210-based probes in *S cerevisiae* (*Pranke et al., 2011*). However, the yeast homolog of GMAP-210 (Rud3p; [*Gillingham et al., 2004*]) does not contain an ALPS motif and we considered the possibility that GMAP-210 had acquired additional interactions in its natural context.

*Figure 5C* shows a cell co-expressing a green construct containing the inverted ALPS motif (invALPS-ACC2-GFP) and a red construct containing the normal ALPS motif (ALPS-ACC1-mCherry). The two constructs had different coiled-coil regions to prevent the formation of heterodimers (*Horchani et al., 2014*). Remarkably, ALPS-ACC1-mCherry and invALPS-ACC2-GFP showed exactly the same Golgi localization despite the fact that these constructs were different along their entire sequence. The Pearson's coefficient between the green and red channels was not only very high (0.75), but also similar to that observed when two coiled-coil constructs harbored an intact ALPS motif (*Figure 5D*). These results demonstrate that the strict order of amino acids in the ALPS sequence is not critical for its fine targeting to the Golgi, making the involvement of stereospecific interactions unlikely.

## The ALPS motif of GMAP-210 interacts with Golgi vesicles as an amphipathic helix

Having ruled out a stereospecific mode of membrane recognition, we wished to test the alternative hypothesis, namely that the ALPS motif of GMAP-210 targets Golgi vesicles by merely adsorbing to their surface as an amphipathic helix. For this, we used a different mutagenesis strategy. Instead of introducing a large number of mutations while keeping the physicochemical properties of the sequence intact, we introduced a minimal number of mutations to disrupt the amphipathic character of the sequence. This included placing a single negatively charged residue in the hydrophobic face (L12D), inserting two alanines to break the regular spacing of polar and non-polar residues



**Figure 5.** Selective targeting of the ALPS motif of GMAP-210 to *cis* Golgi vesicles without stereospecific interactions. (**A**, **B**) Sequence comparison of the ALPS motif of GMAP-210 and the corresponding inverted motif (invALPS). The two sequences have low identity (≈ 25%; **A**), but helical wheel-representations (**B**) show that they display the same amphipathic character. Helical representations were made with Heliquest (*Gautier et al., 2008*).(**C**) A coiled-coil construct harboring an inverted ALPS motif (invALPS-ACC2-GFP) displays the same subcellular localization as a construct harboring the original ALPS motif (ALPS-ACC1-mCherry). (**D**) Pearson coefficient between the green and red channels for the constructs shown in (**C**) as well as for other combinations of constructs or endogenous proteins. 30 cells were examined from 3 independent experiments. The vertical black bars show the mean ± SD. All images were acquired with a confocal microscope and were analyzed using Volocity software. Scale bars: 10 μm, Z plane.

(InsAA20), and replacing a glycine by a proline to create a bend (G11P). In all cases, the *cis*-Golgi localization of the mutated constructs dropped in favor of the cytosolic fraction (*Figure 6A,C* and *Figure 6—figure supplement 1*).

The ALPS motif of GMAP-210 displays a monotonous sequence made of similar helical turns (typically [LGx] or [LGxx]; *Figure 5A*). We reasoned that if all these turns insert at the surface of Golgi vesicles, their contribution to the overall binding energy should be similar. We thus designed a series of truncated forms in which two helical turns were successively deleted. All truncated constructs showed strong impairment in Golgi targeting suggesting that the full ALPS sequence contributes to membrane binding (*Figure 6B,C* and *Figure 6—figure supplement 1*).

In conclusion, the mutagenesis analysis shown in *Figures 5* and *6* suggests that, despite being very selective for Golgi vesicles, the ALPS motif of GMAP-210 does not target them through stereo-specific interactions, but simply through its amphipathic properties.

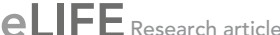

**Figure 6.** The ALPS motif of GMAP-210 interacts with Golgi vesicles as an amphipathic helix. (**A**) Effect of mutations disrupting the amphipathic character of the ALPS motif of GMAP-210. For each mutant, a helical plot made with Heliquest (*Gautier et al., 2008*) is shown. Mutation L12D introduces a negative charge in the hydrophobic face. Ins20AA corresponds to 2 alanines inserted in the middle of the ALPS sequence. G11P should rigidify the sequence. (**B**) Effect of gradual truncation of the ALPS sequence. All mutants shown in (**A**) and (**B**) were derived from ALPS-ACC1-GFP. (**C**) Quantification. For both (**A**) and (**B**), ten cells were analyzed for each mutant. Once the background was subtracted, two ROI with same area were applied in the Golgi and in the cytosol. The average fluorescence was determined in each ROI and the Golgi/cytosol ratio was then calculated. All images were acquired with a confocal microscope and were analyzed using Volocity software. Scale bars: 10 μm, Z projections. Scale bars: 10 μm. The figure shows one experiment where all constructed were transfected in parallel and ten cells were analyzed for each condition. The horizontal bars show the means. Other experiments were performed with subgroups of mutants and gave similar results.

The following figure supplement is available for figure 6:

**Figure supplement 1.** Effects of point mutations on the Golgi localization of the ALPS motif of GMAP-210.

## The sparse distribution of hydrophobic residues in ALPS enables selective Golgi targeting

ALPS motifs are defined by the abundance of uncharged residues (S, T, G) in their polar face and contain bulky hydrophobic residues (F, L, W) in their non-polar face (*Bigay et al., 2005*; *Mesmin et al., 2007*; *Drin et al., 2007*). As compared to ALPS, the most contrasting amphipathic α-helix is α-synuclein, which contains charged residues (K, E) in its polar face and small hydrophobic residues (V, T) in its non-polar face. We previously showed that the opposite traits of ALPS and α-synuclein correlate with their different subcellular localization (*Pranke et al., 2011*). However, the exquisite subcellular localization of the ALPS motif of GMAP-210 remains puzzling when compared to that of other amphipathic α-helices. The N-terminal helices of Arf1 and Sar1, two small G proteins that drive the assembly of protein coats on the ER and the Golgi, are also poor in charged residues (*Drin and Antonny, 2010*). Yet, these small G proteins display a wider cellular distribution than the ALPS motif of GMAP-210. Moreover, in biochemical assays, ALPS-motif containing proteins are much more sensitive to membrane curvature than Arf1 (*Bigay et al., 2003*; *Drin et al., 2008*; *Ambroggio et al., 2010*). Therefore, we surmised that another feature of the ALPS motif might contribute to its selective membrane adsorption properties.

Sequence comparison shows that the distribution of hydrophobic residues in the ALPS motif of GMAP-210 strongly differs from that of Arf1 and Sar1 (*Figure 7A*). In ALPS, most helical turns (9 out of 11) contain a single hydrophobic residue (e.g. LxxxLxxxL); in Arf1 and Sar1, most helical turns contain two adjacent hydrophobic residues (e.g. IFxxLFxxLF in Arf1). To determine whether the sparse distribution of hydrophobic residues in ALPS plays a role in membrane selectivity, we created a condensed ALPS motif (condALPS) by deleting polar residues such as to create hydrophobic pairs akin to what is found in Arf or Sar (*Figure 7A*). The resulting amphipathic helix was shortened but retained all hydrophobic residues of the wild-type form.

*Figure 7B* compares the cellular localization of coiled-coil constructs (ACC1-GFP) harboring an intact ALPS motif, a condensed ALPS motif, or the Sar1 amphipathic helix (for other examples see also *Figure 7—figure supplements 1* and *2*). Whereas, the former was concentrated in the *cis* Golgi area, the constructs harboring condALPS or Sar1AH decorated not only the Golgi but also the nuclear envelope, the ER network and lipid droplets. Thus, amphipathic helices containing hydrophobic doublets seem more promiscuous than those containing sparse hydrophobic residues. To confirm this finding, we added additional hydrophobic residues in the ALPS motif of GMAP-210 or in shorter constructs (*Figure 7C* and *Figure 7—figure supplement 2*). Again, we observed that the mutated constructs localized to various organelles including the Golgi, the ER and lipid droplets. We also noticed that the only truncated mutant of ALPS that showed significant Golgi localization ([1–6/29–38]) was less typical in sequence as it contained the only helical turn with two adjacent hydrophobic residues ($_4$WL$_5$; *Figure 6B,C*). This observation further confirms the importance of the distribution of hydrophobic residues for tuning the avidity and selectivity of amphipathic helices for lipid membranes.

## A 'condensed' ALPS mutant binds lipid membranes regardless of curvature in vitro

Next, we wished to compare the biochemical properties of coiled-coil constructs harboring condALPS or ALPS. All constructs containing condALPS were poorly soluble when expressed in bacteria, thereby preventing an analysis similar to that shown in *Figure 1F*. To circumvent this limitation, we used two synthetic peptides corresponding to the ALPS and condALPS sequences, respectively, and followed their partitioning to model membranes by monitoring the blue shift and increase in tryptophan fluorescence that occur upon membrane binding (*Drin et al., 2007*). In the case of wild-type ALPS, membrane binding to liposomes was very low, except for the smallest liposomes obtained by sonication (*Figure 7D*). In contrast, condALPS partitioned readily to all liposomes regardless of their size (*Figure 7D*). Thus, the sparse distribution of hydrophobic residues in ALPS motif contributes to its sharp sensitivity to membrane curvature.

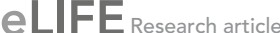

**Figure 7.** The sparse distribution of hydrophobic residue in ALPS determines its selective cellular targeting and its sensitivity to membrane curvature. (**A**) Sequence comparison between the ALPS motif of GMAP-210 and the amphipathic helices of Arf1 and Sar1. Hydrophobic residues are shown in

*Figure 7 continued on next page*

*Figure 7 continued*

bold. In ALPS, most hydrophobic residues are sparse. In Arf1 and Sar, most hydrophobic residues are paired. By removing polar residues between sparse hydrophobic residues, the ALPS sequence was condensed (condALPS) to create hydrophobic pairs. (B) Localization of ALPS-ACC1-GFP, condALPS-ACC1-GFP, and Sar1-ACC1-GFP in RPE1 cells. Lipid droplets were revealed by lipidTox staining (red). Imaged were acquired from living cells using a spinning disk microscope. Z planes, scale bar: 10 µm. (C) Effect of adding hydrophobic residues in the ALPS 1–38 sequence or in the 8–30 truncated mutant. All mutants derived from the ALPS-ACC1-GFP construct. (D) Sensitivity of ALPS and condALPS peptides to membrane curvature. The peptides (1 µM) were incubated with liposomes (lipids 0.5 mM) of defined size. Tryptophan fluorescence was measured between 300 to 450 nm. The right plot shows the relative fluorescence at 340 nm as a function of liposome radius, setting the fluorescence in solution at 1. Error bars: SEM from four independent experiments. Scale bars: 10 µm.

The following figure supplements are available for figure 7:

**Figure supplement 1.** Subcellular localization of ALPS-ACC1-GFP, condALPS-ACC1-GFP, and Sar1-ACC1-GFP in RPE1 and U2OS cells.

**Figure supplement 2.** The sparse distribution of hydrophobic residue in ALPS determines its selective cellular targeting.

## Lipid-packing defects are maximal in vesicles with an ER/*cis* Golgi-like composition

The mechanism by which amphipathic helices adsorb on the surface of lipid membranes is difficult to experimentally address (*Hristova et al., 1999*). In the case of ALPS motifs, molecular dynamics (MD) simulations suggest a plausible sequence whereby bulky hydrophobic residues randomly insert into lipid packing defects (i.e. voids) present on the membrane interfacial region (*Vamparys et al., 2013*; *Vanni et al., 2013*). When the density of these defects is high enough, multiple hydrophobic insertions occur, thereby favoring ALPS folding at the membrane surface. Importantly, a topographic analysis indicates that the abundance of lipid packing defects that are large enough to accommodate bulky hydrophobic residues sharply increases with both membrane curvature and with lipid unsaturation in agreement with the adsorption properties of ALPS motifs (*Vanni et al., 2014*). However, these simulations were performed with simple PC membranes.

We wished to determine if lipid-packing defects of the same size are present in vesicles travelling between the ER and *cis* Golgi (i.e. formed by the COPI or COPII coats). For this, we built coarse-grained models of bilayers of different shape with a composition approaching that of the ER/*cis* Golgi interface (*van Meer et al., 2008*) and including COPI vesicles for which a comprehensive lipid analysis exists (*Brügger et al., 2000*) (*Figure 8A*). The models consisted in PC (60 mol%) and phosphatidylethanolamine (PE; 30 mol%) as the main glycerophospholipid lipids as well as 10 mol% cholesterol (*Figure 8B*). The acyl chain distribution of PC and PE matched that of COPI vesicles with a minor proportion of saturated species (C16:0-C16:0), an intermediate level of monounsaturated species (C18:1-C18:1), and a majority of mixed saturated-monounsaturated species (C16:0-C18:1) (*Figure 8B*).

By scanning the surface of these bilayers, we determined the distribution of lipid packing defects. As observed previously (*Cui et al., 2011*; *Vamparys et al., 2013*; *Vanni et al., 2014*), the abundance of lipid packing defects falls off with their area according to an exponential decay $e^{(-A/Ac)}$, where *A* is the area of the defect and *Ac* is an area constant (*Figure 8C*). Thus, large defects are always less abundant than smaller ones and a high *Ac* value indicates a bilayer rich in lipid packing defects. The analysis shows that *cis* Golgi bilayers fall between POPC and DOPC membranes in term of distribution of lipid packing defects and that membrane curvature increases the probability of having large defects (*Figure 8C*). To quantify this increase, we calculated the relative abundance of lipid packing defects between curved and flat Golgi bilayers (*Figure 8C*, inset). Lipid packing defects large enough to accommodate ALPS bulky hydrophobic residues (>20 Å$^2$; [*Vanni et al., 2013*]) were *ca.* 2–3 times more abundant in tubules and 3–5 times more abundant in vesicles as compared to flat bilayers. We concluded that the ALPS motif is best adapted to sense the surface of the most highly curved membranes of the early secretory pathway, notably vesicles formed by the COPI and COPII coats and whose radius ranges between 30 and 40 nm (*Malhotra et al., 1989*; *Barlowe, 1994*).

**Figure 8.** Distribution of lipid packing defects in model *cis* Golgi membranes. (**A**) Typical coarse grained models of lipid bilayers with a lipid composition similar to that of COPI vesicles (*Brügger et al., 2000*) and with a flat, tubular or spherical geometry. (**B**) Lipid composition of the bilayers. (**C**) Lipid packing defect distribution. For comparison, the results from simulations with flat POPC or spherical DOPC membranes (*Vanni et al., 2014*) are also shown. The inset shows the ratio between the number of defects in the vesicle or in the tube as compared to the flat membrane.

## Discussion

The Kobayashi and Pagano experiment was remarkable in its simplicity – mixing liposomes with perforated cells – and in its outcomes: the Golgi has an exceptional liposome tethering capacity, which is accompanied by size filtration (*Kobayashi and Pagano, 1988*). By connecting these pioneering observations with the molecular and cellular properties of the golgin GMAP-210 (*Drin et al., 2008*; *Wong and Munro, 2014*; *Sato et al., 2015*; *Roboti et al., 2015*), our study suggests that the ALPS

motif of GMAP-210 acts a filter at the *cis* entrance of the Golgi to select transport vesicles according to their size and bulk lipid composition.

The ALPS motif of GMAP-210 should be one of the first binding sites that a vesicle entering the golgin matrix should encounter and its sharp preference for small liposomes (R < 40 nm) suggested a mechanism for vesicle capture (*Drin et al., 2007*, *2008*). The present study with perforated cells further underlines the relevance of this mechanism in a cellular context and also gives insights about its efficiency. We estimate that the Golgi can capture hundreds of small DOPC liposomes (R ≈ 30 nm). This range compares reasonably well with the vesicle density at the Golgi as determined by electron tomography: Ladinsky et al. reported *ca.* 400 vesicles in a tomogram corresponding to 5% of the Golgi area, whereas Marsh et al. reported >2000 vesicles in a larger region (*Ladinsky et al., 1999*; *Marsh et al., 2001*). Other clues for the efficiency of this mechanism are the massive accumulation of small (R ≈ 30 nm) vesicles in the form of tight clusters when GMAP-210 is overexpressed in yeast or in mammalian cells (*Pernet-Gallay et al., 2002*; *Pranke et al., 2011*) as well as the displacement of the liposome capture properties when cells express GMAP-210 at the surface of mitochondria (*Figure 3*). However, GMAP-210 might not be the only golgin accounting for the ability of the Golgi to capture liposomes. Among the various golgins present in mammalian cells (*Gillingham and Munro, 2016*), we identified another ALPS motif in giantin (*Drin et al., 2007*). This motif is located within the coiled coil region and therefore might not be involved in the initial capture of vesicles but rather in their movement within the golgin matrix. In the future, combining genetic tools to modify the repertoire of vesicle tethers and liposome-binding experiments on perforated cells should facilitate the analysis of the various modes of vesicle capture by the Golgi and other organelles (*Ho and Stroupe, 2015*; *Ohya et al., 2009*; *Kuhlee et al., 2015*).

Mitochondria de-routing experiments indicate that GMAP-210 is selective for vesicles of the early secretory pathway (*Wong and Munro, 2014*). Furthermore, functional and structural dissection of GMAP-210 indicates that its ALPS motif has an essential role in vesicle capture (*Sato et al., 2015*). Our study suggests that the vesicle selectivity of GMAP-210 does not arise from stereospecific interactions between the ALPS motif and particular protein or lipid determinants, but from the unusual amphipathic properties of this motif. One striking observation is that the targeting of GMAP-210 ALPS to the Golgi is resistant to sequence inversion but is very sensitive to mutations that modify its amphipathic character (*Figures 5*, *6*). Notably, mutations that disrupt the helix amphipathy cause ALPS dissociation, whereas mutations that reinforce the hydrophobic face promote recognition of other organelles besides the Golgi. Thus, the physicochemical properties of the ALPS motif of GMAP-210 seem exquisitely tuned for selective adsorption on the surface of small vesicles of the early secretory pathway. Besides the chemistry of ALPS per se, the dimeric architecture of GMAP-210 also contributes to membrane curvature recognition (*Doucet et al., 2015*).

ALPS motifs differ from most membrane-adsorbing helices by the lack of charged residues and the abundance of small neutral residues in the polar face (*Drin et al., 2007*; *Drin and Antonny, 2010*; *Pranke et al., 2011*). The present study identifies the distribution of hydrophobic residues as a second parameter accounting for the hypersensitivity of ALPS motifs to membrane curvature (*Figure 6*). In ALPS, most helical turns contain a single hydrophobic (H) residue (HxxHxxH...) (*González-Rubio et al., 2011*). In Arf1 and Arf1-like proteins, the N-terminal amphipathic helix contains two hydrophobic residues per turn (HHxxHHx...). Condensing the ALPS sequence in order to pair all single hydrophobic residues makes the helix no longer sensitive to membrane curvature and eliminates its restrictive Golgi distribution (*Figure 6*). We suggest that the presence of a single hydrophobic residue per turn prevents each helical turn from being too tightly bound. Since this feature is repeated many times, this amplifies the helix sensitivity to the lipid packing properties of the membrane interface (*Antonny, 2011*). In conclusion, the ALPS motif of GMAP-210 is optimized to select small vesicles, whereas Arf1 and Arf-like proteins are more permissive to membrane curvature, which enables them to initiate, by a GTP switch, the recruitment of various effectors (coats, tethers, lipid-modifying enzymes, lipid transporters) on membranes of diverse shape (*Cherfils, 2014*).

The membrane of the ER and the *cis* Golgi differ from that of other organelles in their bulk features, including low sterol, low electrostatics and high level of unsaturation (*van Meer et al., 2008*; *Bigay and Antonny, 2012*). Molecular dynamics simulations suggest that this latter trait, together with the high curvature of transport vesicles (R = 30–40 nm, for both COPI and COPII vesicles; [*Malhotra et al., 1989*; *Barlowe, 1994*]) leads to a high level of lipid packing defects, which enable the adsorption of ALPS motifs (*Figure 8* and [*Vanni et al., 2013*]). The observation that excess

phospholipid saturation prevents the correct partitioning of ArfGAP1 at the Golgi supports this model (*Vanni et al., 2014*). In addition, COPI vesicles, which bud from the *cis* Golgi, are rich in unsaturated PC and poor in cholesterol and saturated lipids (*Brügger et al., 2000*). Therefore, we propose that GMAP-210 recognizes vesicles that shuttle cargoes between the ER and the *cis* Golgi because their membrane combines the highest level of unsaturation with the highest level of curvature. In contrast, small vesicles on the *trans* side of the Golgi are probably ignored by GMAP-210 because their physical features, notably a more ordered bilayer structure (*Klemm et al., 2009*), are not adapted to the ALPS motif.

In conclusion, GMAP-210 seems to act as a general filter at the *cis* side of the Golgi to select vesicles solely on the basis of their small size and loose lipid packing structure. This mechanism might be sufficient to sort pre Golgi *versus* post Golgi vesicles, thereby maintaining the polarity of this organelle, because changes in lipid composition along the secretory pathway (e.g. sterol accumulation and sphingolipid synthesis) occur mostly at the *trans* side (*van Meer et al., 2008*; *Bigay and Antonny, 2012*). In contrast, GMAP-210 is probably not able to distinguish subclasses of vesicles at the ER/cis Golgi interface. GMAP-210 controls both anterograde and retrograde flows between the ER and the cis Golgi (*Pernet-Gallay et al., 2002*; *Roboti et al., 2015*) and, when artificially anchored on the mitochondria, captures ER-derived vesicles as well as those that contain Golgi resident enzymes (*Wong and Munro, 2014*). This lack of discrimination might be advantageous for building a new *cis* Golgi cisternae by merging these different vesicles. The only transport intermediates at the ER/Golgi interface that should not be captured by GMAP-210 are large carriers, which transport proteins too big to be packaged into classical vesicles (*Malhotra and Erlmann, 2015*). In this respect, it is interesting to note that deletion of GMAP-210 in mouse leads to defects in the secretion of a relatively small matrix protein, perlecan, but not of large collagens (*Smits et al., 2010*). GMAP-210 is also reported to function in the cilium formation (*Follit et al., 2008*). A plausible hypothesis is that GMAP-210 diverts vesicles with a special level of lipid unsaturation from the Golgi. As such, GMAP-210 could help building an organelle that, despite being continuous with the plasma membrane, displays a very different lipid composition (*Verhey and Yang, 2016*).

## Materials and methods

### Constructs

ACC1 and ACC2 are artificial coiled-coils. Their design, their sequences and the corresponding synthetic genes (Eurofins, Luxembourg) have been previously described (*Horchani et al., 2014*). The ALPS motif of GMAP-210 (aa 1–38) was cloned in a pEGFP-N1 vector (Clonetech, Japan) upstream of ACC1, thereby giving the ALPS-ACC1-GFP construct. The cloning restriction site was BamHI. The ALPS point mutants (L12D, G11P, Ins20AA), the ALPS truncated mutants ([1–14], [8–38], [15–38], [1/6–29/38], [22–38]), as well as condensed ALPS were prepared from ALPS-ACC1-GFP using the QuikChange Lightning kit (Agilent Technologies, CA., USA) according to the manufacturer's instructions. InvALPS was prepared from a synthetic gene including the ACC2 sequence. Sar1-ACC1-GFP was prepared by inserting the amphipathic helix of Sar1 in the ACC1-GFP vector using directed mutagenesis (*Doucet et al., 2015*).

For ALPS-ACC1-GFP-FKBP, we inserted ALPS-ACC1-GFP in a ProteoTuner vector (Clonetech, Japan) that contains a modified FKBP domain (degradation domain). This domain was mutated back to the wild-type form by introducing the following mutations: G31E, V36F, G71Q and E105K. Mito-FRB was a gift from M. Lowe (*Sato et al., 2015*).

### Cell culture, protein expression and antibodies

hTERT-RPE1 cells (purchased from ATCC, VA, USA) were cultured in DMEM/F12 medium with gluta-MAX (Gibco, Thermo Fisher Scientific, MA, USA) containing 10% serum, 1% antibiotics (Zell Shield, Minerva Biolabs, Germany) and were incubated in a 5% $CO_2$ humidified atmosphere at 37°C. hTERT-RPE1 cells stably expressing Rab6-GFP (gift from B Goud) or ArfGAP1(R50K) (*Vanni et al., 2014*) were cultured in the same medium but supplemented with a geneticin (G418 500 µg/ml). In brief, cells were seeded on glass coverslips previously coated with fibronectin (4 µg/ml). Plasmids were transfected using Lipofectamine 2000 (Invitrogen, CA, USA). After 3–5 hrs, the cells were fixed with PFA 3% for 20 min at room temperature. Golgi staining was performed using mouse monoclonal

antibody against GM130 (ref 610822-BD, Transduction laboratories, KY, USA) and sheep polyclonal antibody against TGN46 (ref AHP500G-AbD, Serotec, Bio-Rad Laboratories, CA, USA). Antibody against endogenous GMAP-210 was provided by Michel Bornens (Institut Curie Paris) (*Infante et al., 1999*). Purified mouse anti-GMAP-210 was from BD transduction laboratories (ref 611712). Coverslips were mounted in Mowiol and observed using a confocal microscope (Leica-SP5 or Zeiss LSM-780). Images were analyzed by volocity software or Image J.

For lipid droplets staining, live cells were incubated for 10 min with HCS lipidTOXred neutral lipid stains (Invitrogen). Experiments with live cells were performed using a spinning disk confocal microscope (UltraVIEW - Perkin Elmer, MA, USA).

## Cell perforation and incubation with liposomes

We followed the method previously described (*Simons and Virta, 1987*; *Kobayashi and Pagano, 1988*). hTERT-RPE1 cells or RPE1-Rab6GFP cells were seeded on glass coverslips (diameter 35 mm) previously coated with fibronectin (10 µg/ml) until confluence. Cells on coverslips were washed in DMEM/F12 medium without red phenol (Gibco Thermo Fisher Scientific, MA, USA) and the excess moisture was removed by touching the edge of the coverslip with a filter paper. A 25 mm diameter filter (HATF, 0.45 µm pore size, Millipore) was soaked in perforation buffer (25 mM HEPES-KOH pH 7.2, 115 mM KCl, 2.5 mM MgCl$_2$), blotted against filter paper, and placed on top of the coverslip. After 1 min at room temperature, the filter was slowly peeled from the cells. The coverslip was placed in a support for life microscopy, and 400 µl of liposomes solution was immediately added. After 5 min of incubation at 30°C, cells were washed several times with DMEM/F12 medium without red phenol (Gibco) and cells were observed in a wide-field microscope (Olympus IX83). The Golgi area was determined by the Rab6-GFP signal, and a corresponding mask was then applied to the red channel to determine the liposome fluorescence in the same area.

## RNA interference

Cells were transfected with 15 nM siRNA using Lipofectamine RNAiMAX (Invitrogen, CA, USA) for a 35 mm glass coverslip. ON-TARGET plus siRNAs targeting GMAP-210 were from Dharmacon (GE Healthcare, UK). They correspond to the siRNAs used by Roboti et al. (*Roboti et al., 2015*) and have the following sequences:

siGMAP-210#1: GGAGAUAGCAUCAUCAGUA,ref:J-012684-05-0002
siGMAP-210#2: CAAGAACAGUUGAAUGUAG, ref:J-012684-06-0002
non-targeting siRNA: UGGUUUACAUGUUGUGUGA, ref:D-001810-02-05

After 72 hr of transfection, cells were used for western blotting or for perforation. Expression of GMAP-210 was revealed with a purified mouse anti-GMAP-210 antibody (BD transduction laboratories, KY, USA) and the loading control was revealed with a monoclonal mouse anti-GFP antibody (Thermo Scientific, MA, USA).

## Liposome-rerouting at mitochondria

Cells coexpressing ALPS-ACC1-GFP-FKBP and Mito-FRB were treated with 1 µM rapamycin (Sigma, MO, USA, ref R8781) for 1 hr to induce targeting of ALPS-ACC1-GFP-FKBP to the mitochondria outer membrane. When indicated, cells were also treated with nocodazole (2.5 µg/ml) to scatter the Golgi into mini stacks. This treatment was done 1 hr before and during rapamycin treatment. Thereafter, the cells were perforated and incubated with liposomes (250 µM lipids) as described above.

## Peptides

The ALPS of GMAP-210 (aa 1–38) peptide and the condensed ALPS peptide were synthesized by ProteoGenix, France:

ALPS GMAP-210: CSS**WL**GGL**G**SG**L**GQS**L**GQ**V**GGS**L**AS**L**TGQ**I**SN**F**TKD**ML** (38 aa)
CondALPS: CSS**WL**GG**LL**GS**LV**GS**LL**T**IF**KD**ML** (24 aa)

Note that the first methionine was mutated into a cysteine and that the sequences contain one tryptophan residue, thereby allowing determination of peptide concentration by absorbance at 280 nm and liposome partitioning by fluorescence measurement.

Due to its high hydrophobicity, condALPS was only soluble in urea. Stock solutions of the two peptides (50 µM) were thus prepared in 4 M urea supplemented with 2 mM DTT.

## Liposomes

All lipids were purchased from Avanti Polar Lipids (AL, USA). Lipids in chloroform were mixed in a pear-shape vial and dried in a rotary evaporator. The lipid film was hydrated at 2 to 10 mM lipids either in HK buffer (Hepes 50 mM, KAcetate 120 mM, pH 7,4) for biochemical experiments or in Hepes 25 mM, KCl 115 mM, MgCl$_2$ 2.5 mM, pH 7,2 for the experiments with perforated cells. After five cycles of freezing and thawing, the liposomes were extruded sequentially through polycarbonate filters with pore diameters of 200, 100, 50 and 30 nm. Alternatively, small liposomes were obtained by sonication with a titanium tip sonicator (2 × 10 min in ice) and centrifuged for 20 min at 45,000 rpm to remove titanium debris. The liposome hydrodynamic radius (Rh) was measured by dynamic light scattering in a Dyna Pro instrument. Liposomes were stored at room temperature and used within one day.

## Calibration of the liposome fluorescence signal at the Golgi

To convert the red fluorescence signal at the Golgi into a number of rhodamine-labeled liposomes, we imaged liposome suspensions in a Malassez cell (thickness 200 μm) using the same settings as that used for cell imaging (oil objective 60X, numerical aperture *NA* = 1.42). The mean fluorescence intensity per pixel increased linearly with liposome concentration (from 0 to 100 μM lipids) giving a conversion factor, *k*:

$k \approx 100\ FU.\mu M^{-1}$

The mean fluorescence intensity per pixel at the Golgi after incubation with small DOPC liposomes is in the range of 2000 to 5000 *FU* (**Figure 1D**), thus corresponding to a lipid concentration *C* = 20 to 50 μM.

From our images, we estimated that the Golgi region occupies a volume in the range of:

$V = 10 \times 10 \times 3 = 300\ \mu m^3 = 3\ 10^{-13}$ L.

In such a volume, a concentration of 20 to 50 μM corresponds to:

$C \times V \times N_{av} = 3.6\ 10^6$ to $9\ 10^6$ lipids, where $N_{av}$ is the Avogadro number.

The number of lipids in a small liposome (*R* = 30 nm) is $\approx 2 \times \frac{4\pi R^2}{s} \approx 3\ 10^4$, where *s* is the surface of one lipid (*s* $\approx$ 0.7 nm$^2$).

Thus, the Golgi has captured 120 to 300 small liposomes.

## NBD and tryptophan fluorescence assay

NBD experiments were performed at 37°C in a 4 × 4 mm cuvette (volume 240 μl) in a FP 8300 fluorimeter (Jasco, Japan). Excitation was set at 505 nm (bandwidth 2.5 nm). Emission was recorded from 520 to 650 nm (bandwidth 5 nm). The cuvette initially contained liposomes of defined size and composition (0.15 mM lipids) in HKM buffer (Hepes 50 mM, pH 7.4, Kacetate 120 mM, MgCl$_2$ 1 mM, DTT 1 mM) and a blank spectrum was recorded. Thereafter, 0.125 μM NBD-labeled GMAP (1–375 fragment with the M1C mutation for NBD labeling [*Pranke et al., 2011*]) was added and another spectrum was recorded. The second spectrum was corrected for the blank.

For tryptophan fluorescence, excitation was set at 280 nm (bandwidth 5 nm) and emission was recorded from 300 to 450 nm (bandwidth 5 nm). The cuvette initially contained liposomes (DOPC/DOPE/Cholesterol 60/30/10 mol/mol; 0.5 mM lipids) of the indicated size in HKM buffer and a blank spectrum was recorded. Thereafter, 1 μM peptide (ALPS or condALPS) was added from 50X stock solutions in urea (4 M) and another spectrum was recorded and corrected for the blank.

## MD simulations

Simulations were done using the MARTINI force field of coarse-grained models of lipids as described previously (*Risselada and Marrink, 2009*; *Vanni et al., 2014*) but using a composite lipid composition similar to that described by Brügger et al. on COPI vesicles (*Brügger et al., 2000*). The bilayer contained 60 mol% PC, 30 mol% PE and 10 mol% cholesterol. The acyl chain profile of PC and PE was 6 mol% C16:0-C16:0, 59 mol% 16:0-C18:1 and 35 mol% C18:1-C18:1. Note that the MARTINI coarse-grained model does not distinguish close lipid species such as C18:0-C18:1 and C16:0-C18:1. Therefore, these two lipid species, which are present in COPI vesicles, were simulated using the same coarse-grained model. The flat, tubular and spherical models of *cis* Golgi-like membranes were built from the corresponding DOPC structures (*Vanni et al., 2014*). In successive steps, the correct percentage of DOPC molecules was changed *via* molecular replacement into the new lipids

(DPPE, POPE, DOPE, DPPC, POPC and cholesterol). After each molecular replacement, the system was minimized using a steepest descent algorithm. Next, the systems were subjected to MD runs of 900 ns (flat, tubule) and 150 ns (vesicle) in the NPT ensemble. Analyses were carried out on the last 600 ns (flat), 450 ns (tubule) and 100 ns (vesicle). All simulations details as well as analysis for lipid-packing defects are identical to those described previously (*Vanni et al., 2014*). Data for pure POPC and DOPC bilayers discussed in the text and shown in *Figure 8C* are also taken from our previous study (*Vanni et al., 2014*).

## Acknowledgements

We thank all lab members for discussions and encouragements and Guillaume Drin, Cathy Jackson, Christine Doucet and Alenka Copic for comments on the manuscript. This work was supported by the ANR (« Investments for the Future » LABEX SIGNALIFE: program reference # ANR-11-LABX-0028-01), the ERC (advanced grant ERC 268 888) and the conseil général des Alpes Maritimes (programme santé CG06). We also thank Martin Lowe for constructs, Bruno Goud for the stable cell lines expressing Rab6-GFP, Jean-François Casella for some initial constructs, Frédéric Brau for support in light microscopy, and the staff of the Centre Commun de Microscopie Appliquée for help with electron microscopy.

## Additional information

### Funding

| Funder | Grant reference number | Author |
|---|---|---|
| Agence Nationale de la Recherche | ANR-11-LABX-0028-01 | Bruno Antonny |
| European Research Council | Advanced Grant 268 888 | Bruno Antonny |
| Conseil Départemental des Alpes Maritimes | appel à projet santé CG06 2013-2014 | Bruno Antonny |

The funders had no role in study design, data collection and interpretation, or the decision to submit the work for publication.

### Author contributions

MM, All experiments except electron microscopy and tryptophan/NBD fluorescence experiments, Conception and design, Analysis and interpretation of data, Drafting or revising the article; RG, Molecular Dynamics Simulations, Acquisition of data, Analysis and interpretation of data; PG, Electron Microscopy, Acquisition of data, Analysis and interpretation of data; HB, Cell biology, Acquisition of data, Analysis and interpretation of data; SV, Molecular Dynamics Simulations and design of ALPS mutants, Analysis and interpretation of data, Drafting or revising the article; BA, Conception and design, Acquisition of data, Analysis and interpretation of data, Drafting or revising the article

### Author ORCIDs

Stefano Vanni, http://orcid.org/0000-0003-2146-1140
Bruno Antonny, http://orcid.org/0000-0002-9166-8668

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
