## [Decision Letter]

Thank you for submitting your article "A filter at the entrance of the Golgi that selects vesicles according to size and bulk lipid composition" for consideration by *eLife*. Your article has been favorably evaluated by Randy Schekman (Senior editor) and three reviewers, one of whom, Suzanne Pfeffer, is a member of our Board of Reviewing Editors.

The reviewers have discussed the reviews with one another and the Reviewing Editor has drafted this decision to help you prepare a revised submission.

This paper follows up on a striking observation from the Pagano lab, that liposomes, especially those of smaller radius, show preferential affinity for the Golgi when added to semi-intact cells. Here, the authors confirm this finding and show that a significant portion of Golgi affinity is due to the ALPS motif of the tethering protein, GMAP210. This study takes a phenomenological observation and converts it into an impactful, cell biological finding with an interesting molecular basis. The authors go on to show that an inverted amphipathic helix is sufficient and a condensed ALPS motif is less compartment-specific. In general the study is carried out to a high standard and the results will be of broad interest to the readers of *eLife*. The reviewers support presentation in *eLife* after the following issues have been adequately addressed.

1) The title of the work is: "A filter at the entrance of the Golgi that selects vesicles according to size and bulk lipid composition." The overall significance of the story would be greatly enhanced if the authors were to demonstrate the physiological significance of their findings in ALPS motif rescue experiments. This is of particular importance given reports suggesting that loss of GMAP210 can cause changes in cilia. The authors should use CRISPR to remove GMAP210 and then rescue with a version of GMAP210 that lacks the ALPS motif. This would allow them to determine if ER to Golgi traffic and intra-Golgi traffic are really dependent on the ALPS motif, and also to confirm the conclusion that the ALPS motif in GMAP-210 is the primary component responsible for liposome capture in permeabilized cells.

2) An unexpected observation is that the ALPS motif alone is targeted to the early Golgi, as if vesicles are accumulating there for interaction. Is this true in GMAP-210 depleted cells? Is this also true for the ALPs motif from ArfGAP1? Please elaborate. Please provide evidence that this really is the Golgi vs. a peri-Golgi compartment.

3) The mitochondrial tethering experiment data in Figure 3 could be better of quality. In Figure 3 it is not clear what are Golgi fragments versus mitochondria. The same applies for Figure 3—figure supplement 1. Additional counterstains for mitochondria and Golgi would clarify the results. Also, some controls seem missing. There should be [no rapamycin+nocodazole], which is the better control for the [rapamycin+nocodazole] samples. Figure 7 and Figure 7—figure supplement 1 also need counter staining for the Golgi. It is not possible to easily discern the Golgi from other membrane compartments without the Golgi counter labelling.

4) Results: "GMAP-210 is responsible (*for a large portion*) of the ability of the Golgi" (please modify). Are there other obvious candidate proteins that can explain the residual fluorescence detected over the Golgi in GMAP-210 depleted cells? Please discuss.

5) Figure 3. Why is nocodazole used? What happens if it is omitted? Also, given that the ALPS motif will capture endogenous small vesicles how can they be sure that the liposomes are not binding to these vesicles rather than directly to the ALPS motif?

6) Figure 6/C. It seems hard to reconcile the small effects seen by immunofluorescence with the large effects shown in the quantitation. This is in contrast with Figure 6 – please explain.

---

## [Author Response]

*[…] The reviewers support presentation in eLife after the following issues have been adequately addressed.*

1) The title of the work is: "A filter at the entrance of the Golgi that selects vesicles according to size and bulk lipid composition." The overall significance of the story would be greatly enhanced if the authors were to demonstrate the physiological significance of their findings in ALPS motif rescue experiments. This is of particular importance given reports suggesting that loss of GMAP210 can cause changes in cilia. The authors should use CRISPR to remove GMAP210 and then rescue with a version of GMAP210 that lacks the ALPS motif. This would allow them to determine if ER to Golgi traffic and intra-Golgi traffic are really dependent on the ALPS motif, and also to confirm the conclusion that the ALPS motif in GMAP-210 is the primary component responsible for liposome capture in permeabilized cells.

The group of Martin Lowe has addressed this point in two papers that we cite several times in the text:

Roboti, P., Sato, K., & Lowe, M. (2015). The golgin GMAP-210 is required for efficient membrane trafficking in the early secretory pathway. Journal of Cell Science, 128(8), 1595–1606. http://doi.org/10.1242/jcs.166710

Sato, K., Roboti, P., Mironov, A. A., & Lowe, M. (2015). Coupling of vesicle tethering and Rab binding is required for in vivo functionality of the golgin GMAP-210. Molecular Biology of the Cell, 26(3), 537–553. http://doi.org/10.1091/mbc.E14-10-1450

In the first paper, the authors use siRNA against GMAP-210 to assess the involvement of GMAP-210 in several traffic steps between the ER and the Golgi (ER > ERGIC > Golgi and Golgi > ER). Rescue experiments are presented in the second paper, which were conducted with various forms of GMAP-210, including ∆ALSP-GMAP-210.

These studies show that the ALPS motif is necessary for the vesicle tethering activity of GMAP-210 (a complete description of these deletion/rescue experiments is shown in Figure 10 of Sato et al). However, Lowe noticed: “We show here that GMAP-210–mediated tethering in cells is mediated exclusively by the N-terminal ALPS motif. On the face of it, this may appear surprising. Tethering is a specific process, with different golgins able to tether different types of vesicles (Wong and Munro, 2014), yet the ALPS motif binds to membrane lipids”. Our study addresses this issue by showing the importance of curvature and lipid unsaturation for selective vesicle capture.

Of note, the efficiency of GMAP-210 silencing with siRNA is very high (typically, > 95% in the case of Lowe; 85% in our case). Therefore, a CRISPR strategy is not mandatory at this point although this approach will be interesting to better address the physiological function of GMAP-210 in the future (as mentioned in the new final paragraph of the Discussion).

As for the physiological function of GMAP-210, two KO mouse have been presented:

Follit, J. A., Agustin, J. T. S., Xu, F., Jonassen, J. A., Samtani, R., Lo, C. W., & Pazour, G. J. (2008). The Golgin GMAP210/TRIP11 Anchors IFT20 to the Golgi Complex. PLoS Genetics, 4(12), e1000315. http://doi.org/10.1371/journal.pgen.1000315

Follit, J. A., Agustin, J. T. S., Xu, F., Jonassen, J. A., Samtani, R., Lo, C. W., & Pazour, G. J. (2008). The Golgin GMAP210/TRIP11 Anchors IFT20 to the Golgi Complex. PLoS Genetics, 4(12), e1000315. http://doi.org/10.1371/journal.pgen.1000315

In both cases, GMAP-210 deletion leads to embryonic death. Follit observed defects in cilium morphology and function through a decrease in Hedgehog signaling. Smits observed defects in skeletal tissue through decrease in perlican secretion. Although our aim was not to resolve these conflicting studies, our study suggests some hypotheses. First, the defect in perlecan secretion and glycosyslation contrasts with the normal fate of type II collagen and aggregan, which are much bigger matrix proteins. Given the ability of GMAP-210 to select vesicles based on their size, GMAP-210 deficiency should affect only matrix proteins that are incorporated into small classical transport vesicles. Second, the primary cilium has a different lipid composition than the plasma membrane in which it is embedded. Thanks to its ability to select transport vesicles according to their bulk lipid composition, GMAP-210 could help making this atypical membrane structure. These two hypotheses are presented in the new final paragraph of the Discussion.

2) An unexpected observation is that the ALPS motif alone is targeted to the early Golgi, as if vesicles are accumulating there for interaction. Is this true in GMAP-210 depleted cells? Is this also true for the ALPs motif from ArfGAP1? Please elaborate. Please provide evidence that this really is the Golgi vs. a peri-Golgi compartment.

Thank you for this suggestion. To address these points, we have conducted new experiments in which the cells were first siRNA treated for 72 hrs to deplete GMAP-210 and then transiently transfected with probes expressing the ALPS motif. Alternatively, we used cells stably expressing ArfGAP1 (see Vanni et al. 2014 Nat Comm) and again treated the cell with GMAP-210 si RNA. These experiments are presented in a new figure (Figure 4—figure supplement 1). In both cases, the reporter (ArfGAP1 or the ALPS containing probe) remains localized at the Golgi in contrast with liposomes, which no longer accumulate (Figure 2). However, Lowe et al. noticed that depletion of GMAP-210 by siRNA causes vesicles to accumulate at the Golgi at the expense of Golgi cisternae. Probably the vesicles are taken in charge by the other golgins but the flow of vesicle capture and fusion is modified by GMAP-210 deletion (see Figure 2 in Roboti et al). Therefore, there should be enough authentic vesicles for ALPS binding, which is what we observe. These new experiments are presented in a new paragraph of the Results section (subsection “Selective *cis* Golgi targeting of the ALPS motif of GMAP-210”, fourth paragraph).

Since this observation does not allow us to connect the two central observations of our paper (the role of GMAP-210 in liposome capture and the targeting of the ALPS motif to the early Golgi), we performed a different experiment, which is shown in a new Figure (Figure 4—figure supplement 2). We perforated cells that over express the ALPS probe (ALPS-ACC1-GFP). We observed more liposome capture in these cells than in control cells. This observation complements the siRNA experiment (Figure 2) and indicates a correlation between liposome tethering activity of the Golgi and the amount of ALPS motif present. We present these new experiments in the Results section (subsection “Selective *cis* Golgi targeting of the ALPS motif of GMAP-210”, last paragraph.

3) The mitochondrial tethering experiment data in Figure 3 could be better of quality. In Figure 3 it is not clear what are Golgi fragments versus mitochondria. The same applies for Figure 3—figure supplement 1. Additional counterstains for mitochondria and Golgi would clarify the results. Also, some controls seem missing. There should be [no rapamycin+nocodazole], which is the better control for the [rapamycin+nocodazole] samples. Figure 7 and Figure 7—figure supplement 1 also need counter staining for the Golgi. It is not possible to easily discern the Golgi from other membrane compartments without the Golgi counter labelling.

We acknowledge that the previous main figure lacked proper controls and that the use of only two colors (red for liposomes, green for the ALPS construct) made the analysis ambiguous without further staining of membrane compartments. In addition, the supplementary figure was a jumble of different controls.

As suggested by the reviewers, we now focus on two conditions in the main figure (Figure 3): + nocodazole – rapamycin and + nocodazole + rapamycin. In the text, we better explain the advantage of using nocodazole to favor competition between Golgi mini stacks and mitochondria for vesicles capture (following the observation of Wong and Munro and of Sato et al.). In addition, three colors are now used: red: liposomes; green: the ALPS constructs; and violet: GM130 (Figure 3) or MitoTracker (Figure 3—figure supplement 1). A second figure (Figure 3—figure supplement 2) shows the effect of liposome size on liposome capture through mitochondria-targeted ALPS. All figures also contain a small scheme to better explain the experimental conditions.

As for the quality of the images, we hope that the new ones are more satisfactory. However, and as explained in the legend of Figure 3—figure supplement 1, cell perforation clearly damages the mitochondria network, leaving remnants that are less connected and show a round shape.

As for Figure 7 a new supplement figure now shows the counter staining for the Golgi (Figure 7—figure supplement 2).

4) Results: "GMAP-210 is responsible (for a large portion) of the ability of the Golgi" (please modify). Are there other obvious candidate proteins that can explain the residual fluorescence detected over the Golgi in GMAP-210 depleted cells? Please discuss.

We have modified the sentence to temper our conclusion: “Altogether, the experiments shown in Figure 2 indicate that GMAP-210 makes a major contribution to the liposome capture property of the Golgi” and added more information on this point in the Discussion (second paragraph).

Thus, we performed a new bioinformatics search for ALPS-like motifs considering all golgins described by Munro in a recent review. The following list shows the corresponding Uniprot entry numbers:

O60763, P82094, Q7Z5G4, Q8IWJ2, Q8TBA6, Q96CN9, Q08378, Q08379, Q13439, Q13948, Q14789, Q15643, Q92805

As before (Drin et al. 2007), we found an ALPS motif in the golgin giantin (entry: Q14789|GOGB1_HUMAN Golgin subfamily). The corresponding sequence is: (2726)NKGLTAQIQSFGRSMSSLQNS(2746). This motif is localized within the coiled-coil region of this giant protein, about 500 a.a., apart from the transmembrane domain. As such, it might act as a binding site for small vesicles deeper in the golgin matrix as compared to the ALPS motif of GMAP-210, which resides at the N-terminus. However, the ALPS motif of giantin shows a moderate but significant probability to be folded as a coiled-coil. The potential role of this motif is now discussed (Discussion, second paragraph).

5) Figure 3. Why is nocodazole used? What happens if it is omitted? Also, given that the ALPS motif will capture endogenous small vesicles how can they be sure that the liposomes are not binding to these vesicles rather than directly to the ALPS motif?

See response to point 3.

*6) Figure 6/C. It seems hard to reconcile the small effects seen by immunofluorescence with the large effects shown in the quantitation. This is in contrast with Figure 6 – please explain.*

The reader eye focuses on the organelle displaying the highest fluorescence level. For quantification, however, we did only not consider the absolute fluorescence at the Golgi but instead the Golgi/cytosol ratio. For this, we divided the fluorescence in a ‘Golgi box’ by the fluorescence in a ‘cytosol box’ of the same surface. This method gives a fair estimate of the Golgi/cytosol partitioning of the various constructs and allows detecting modest changes between constructs that do decorate the Golgi but show different levels in the cytosol. However, we acknowledge that the image contrast was not optimal to appreciate some differences in the protein level in the cytosol. We have added a supplementary figure in gray levels to better illustrate this point (Figure 6—figure supplement 1).